# A coupled autoencoder approach for multi-modal analysis of cell types

Rohan Gala, Nathan Gouwens, Zizhen Yao, Agata Budzillo, Osnat Penn,
Bosiljka Tasic, Gabe Murphy, Hongkui Zeng, Uygar Sümbül
Allen Institute, Seattle, WA 98109
`rohang@alleninstitute.org, uygars@alleninstitute.org`

## Abstract

Recent developments in high throughput profiling of individual neurons have spurred data driven exploration of the idea that there exist natural groupings of neurons referred to as cell types. The promise of this idea is that the immense complexity of brain circuits can be reduced, and effectively studied by means of interactions between cell types. While clustering of neuron populations based on a particular data modality can be used to define cell types, such definitions are often inconsistent across different characterization modalities. We pose this issue of cross-modal alignment as an optimization problem and develop an approach based on coupled training of autoencoders as a framework for such analyses. We apply this framework to a Patch-seq dataset consisting of transcriptomic and electrophysiological profiles for the same set of neurons to study consistency of representations across modalities, and evaluate cross-modal data prediction ability. We explore the problem where only a subset of neurons is characterized with more than one modality, and demonstrate that representations learned by coupled autoencoders can be used to identify types sampled only by a single modality.

## 1    Introduction

Computation in the brain can involve complicated interactions between millions of different cells. Identifying cell types and their stereotypical interactions based on functional and developmental characteristics of individual cells has the potential to reduce this complexity in service of our efforts to understand the brain. However, capturing the notion of a cell type identity that is consistent across different single cell characterization modalities such as transcriptomics, electrophysiology, morphology, and connectivity has been a challenging computational problem [1, 2, 3, 4, 5].

A general approach to understand correspondence between cell type definitions based on different modalities [3] is to evaluate the degree to which the observable cellular features themselves can be aligned across the modalities. The existence of such alignment would allow one to determine an abstract, potentially low-dimensional representation for each cell. In such a scenario, different transformations could be used to generate realizations of the features measured in the different modalities from the abstract representation itself. Moreover, tasks such as clustering to define cell types could be performed on such representations obtained for cell populations. Here, we propose a method to reveal such abstract identities of cells by casting it as an optimization problem. We demonstrate that (i) cell classes defined by a single data modality can be predicted with high accuracy from observations measuring seemingly very different aspects of neuronal identity, and (ii) the same framework enables cross-modal prediction of raw recordings.

Well known approaches to obtain coordinated representations [6] from multi-modal datasets include the canonical correlation analysis (CCA) and its nonlinear variants [7, 8]. These techniques involve calculation of explicit transformation matrices and possibly parameters of multi-layer perceptrons.

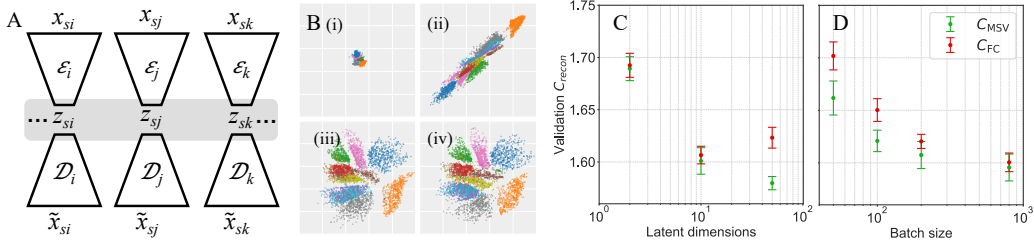

Figure 1: **(A)** Illustration of a $k$-coupled autoencoder. **(B)** 2D representations of the MNIST dataset obtained by one agent of a 2-CAE for various forms of $C_{\text{coupling}}$. Colors represent different digits. (i) Representations shrink to zero in the absence of scaling (Eq.2). (ii) Representations collapse to a line if the scaling is based on batch normalization [11]. Reasonable representations are obtained with $C_{\text{FC}}$ (iii) and $C_{\text{MSV}}$ (iv). $C_{\text{MSV}}$ and $C_{\text{FC}}$ lead to identical $C_{\text{recon}}$ when the full covariance matrix estimates are reliable. For large latent dimensionality **(C)** or small batch sizes **(D)**, $C_{\text{MSV}}$ leads to lower $C_{\text{recon}}$ (mean $\pm$ SE, $n = 10$).

Another recent approach for this problem is the correspondence autoencoder architecture [9], wherein individual agents are standard autoencoders that encode a high dimensional input into a low dimensional latent space from which the input is reconstructed [10]. The trained network is expected to align the representations without any explicit transformation matrices. However, in the absence of any normalization of the representations, the individual agents can arbitrarily scale down their representations to minimize the coupling cost without a penalty on reconstruction accuracy. While Batch Normalization [11] prevents the representations from collapsing to zero by setting the scale for each latent dimension independently, it permits a different pathological solution wherein the representations collapse onto a one dimensional manifold. We present a rigorous analysis of these problems, and show that normalization with the full covariance matrix of the mini-batch is sufficient, as expected [8], to obtain reasonable latent space representations. However, this calculation can be prohibitively inaccurate depending on the latent space dimensionality and batch size ("curse of dimensionality"). Therefore, we propose an alternative normalization that relies only on estimating the minimum eigenvalue of this covariance matrix. Moreover, we derive a probabilistic setting for the cross-modal representation alignment problem and show that our optimization objective can be interpreted as the maximization of a likelihood function, which suggests multiple generalizations of our current implementation.

While there is limited literature on analysis of multi-modal neuronal recordings from a cell types perspective, the advent of large transcriptomic datasets have led to a recent surge of interest in unimodal characterization methods for such data [12, 13, 14, 15, 16, 17]. In particular, Lopez *et al.* [17] propose a generative model for transcriptomic data using variational inference on an autoencoding architecture, and apply $k$-means clustering on the latent representation. While the commonly used Gaussian prior is in contrast with the search for discrete cell classes, mixture model priors [18] are not easily applicable to cases with potentially hundreds of categories. Here, we fit a Gaussian mixture on the latent space representation following the optimization of a discriminative model. We study cross-modal prediction of cell types and raw data with this approach.

Finally, our method can work with partially paired datasets. This setting raises two problems of practical significance for cell type classification: (i) would types that are not sampled by some modalities be falsely aligned to other types? (ii) would types that are sampled by all modalities in the absence of any pairing knowledge have consistent embeddings across the modalities? We demonstrate the utility of our approach in addressing these problems by designing a controlled experiment.

## 2 Theory

### 2.1 Optimization framework

An illustration of the multi-agent autoencoder architecture is shown in Fig. 1A, where agent $i$ receives input $x_{si}$ for which it learns a latent representation $z_{si}$. This representation is used to obtain a reconstruction of the input, $\widetilde{x}_{si}$. The representation learned by a given agent is compared to those learned by all other agents to which it is coupled through a dissimilarity measure. The agents minimize an overall cost function $C$, that consists of penalties on reconstruction error $C_{\text{recon}}$, and mismatches compared to representations learned by other agents, $C_{\text{coupling}}$. The trade-off between learning a representation that minimizes reconstruction error, and one that agrees with the representations learned by other agents is controlled by a coupling constant, $\lambda$.

Formally, we define the $k$-coupled autoencoding tuple ($k$-CAE) $\Phi$ as

$$\Phi = (\{(\mathcal{E}_i, \mathcal{D}_i, r_i)\}_{i \in K}, c, \lambda),$$

where $K$ is an ordered, finite index set, $\mathcal{E}_i, \mathcal{D}_i$ are continuous operators that can express any linear transformation, $\mathrm{codomain}(\mathcal{E}_i) = \mathrm{domain}(\mathcal{D}_j)$, $i, j \in K$, $\lambda \geq 0$, and $r_i$ and $c$ are non-negative convex functions.

For a set of inputs $X = \{(x_{s1}, x_{s2}, \ldots, x_{sk}), s \in S\}$, we define the loss of the $k$-CAE $\Phi$ as

$$C_\Phi(X) = C_{\text{recon},\Phi}(X) + \lambda C_{\text{coupling},\Phi}(X), \tag{1}$$

where

$$C_{\text{recon},\Phi}(X) = \sum_{s \in S} \sum_{i \in K} r_i(x_{si} - \mathcal{D}_i(\mathcal{E}_i(x_{si}))), \quad C_{\text{coupling},\Phi}(X) = \sum_{s \in S} \sum_{\substack{i,j \in K, \\ i < j}} c(\mathcal{E}_i(x_{si}) - \mathcal{E}_j(x_{sj})).$$

In the rest of this paper, we will use the following simplified notation: $C = C_\Phi(X)$, $C_{\text{recon}} = C_{\text{recon},\Phi}(X)$, $C_{\text{coupling}} = C_{\text{coupling},\Phi}(X)$. We will also use the scaled squared Euclidean distance for $r_i$: $r_i(x_{si} - \mathcal{D}_i(\mathcal{E}_i(x_{si}))) = \alpha_i \|x_{si} - \mathcal{D}_i(\mathcal{E}_i(x_{si}))\|_2^2$, $\alpha_i > 0$. When $c$ is also chosen as the squared Euclidean distance and $\alpha_i = 1$ for all $i$, one obtains the cost function of Feng *et al.* [9], $c(\mathcal{E}_i(x_{si}) - \mathcal{E}_j(x_{sj})) = \|\mathcal{E}_i(x_{si}) - \mathcal{E}_j(x_{sj})\|_2^2$:

$$C_{\text{recon}} = \sum_{s \in S} \sum_i \|x_{si} - \widetilde{x}_{si}\|_2^2, \qquad C_{\text{coupling}} = \sum_{s \in S} \sum_{i < j} \|z_{si} - z_{sj}\|_2^2. \tag{2}$$

Here, $z_{si} = \mathcal{E}_i(x_{si})$ and $\widetilde{x}_{si} = \mathcal{D}_i(\mathcal{E}_i(x_{si}))$ denote the latent representation and reconstruction obtained by the $i$-th autoencoder respectively. Subscripts $i$ and $j$ are indices over the individual agents in the coupled architecture. When $k = 2$, these definitions coincide with those proposed by [9] across a set of samples $S$.

The following proposition states that the coupling cost, $C_{\text{coupling}}$ in Eq. 2, can be minimized by scaling the representations by an arbitrarily small value without affecting reconstruction error, $C_{\text{recon}}$. Intuitively, the encoder sub-network of each agent introduces such a scaling to minimize $C_{\text{coupling}}$, and the corresponding decoder sub-network simply inverts this scaling, leaving $C_{\text{recon}}$ unchanged (Fig. 1B(i)).

**Proposition 1.** *Representations of the $k$-CAE that minimize the loss in Eq. 1 with $C_{\text{coupling}} > 0$ satisfy $\|z_{si}\| < \epsilon$, for any norm $\|\cdot\|$, input set $X$, $\epsilon > 0$, and all $s, i$.* (Proof in supp. material)

## 2.2 Scaling latent representation with batch normalization

A way to alleviate the shrinking representation problem is to impose a length scale on the representation. Mini-batch statistics can be used to determine such a scale, as is the case with batch normalization [11]. In its conventional implementation, each dimension $m$ is centered and scaled by empirical estimates of the population mean $\mathbb{E}_s(z_{si}(m))$, and standard deviation $\sigma_s(z_{si}(m))$ based on mini-batch samples:

$$C_{\text{coupling}} = \sum_{s \in S} \sum_{i < j} \|\bar{z}_{si} - \bar{z}_{sj}\|_2^2, \qquad \bar{z}_{si}(m) = \frac{z_{si}(m) - \mathbb{E}_s(z_{si}(m))}{\sigma_s(z_{si}(m))} \tag{3}$$

This, however, permits the agents to collapse their representations to a 1D manifold (Fig. 1B(ii) and Prop. 2). Batch normalization using the full covariance matrix resolves this issue, Fig. 1B(iii) [8]:

$$C_{\text{coupling}} = \sum_{s \in S} \sum_{i < j} \|\hat{z}_{si} - \hat{z}_{sj}\|_2^2, \qquad \hat{z}_{si} = (\mathbf{B}_i^T \mathbf{B}_i)^{-\frac{1}{2}} z_{si} \tag{4}$$

Here $\mathbf{B}_i$ is the $n \times p$ mini-batch matrix where $n$ and $p$ denote mini-batch size and representation dimensionality respectively. Note that $\mathbf{B}_i$ consists of centered representations $z_{si}$ for the mini-batch $S$, scaled by $\sqrt{n-1}$. For reference, the overall cost function in this case is

$$C_\Phi = \sum_{s \in S} \sum_i \alpha_i \|x_{si} - \widetilde{x}_{si}\|_2^2 + \lambda \sum_{i < j} \|\hat{z}_{si} - \hat{z}_{sj}\|_2^2. \tag{5}$$

We now formalize our intuition and the experimental evidence in Fig. 1B. Let $\mu_i = \frac{1}{|S|} \sum_{s \in S} z_{si}$, $V_i = \frac{1}{|S|-1} \sum_{s \in S} (z_{si} - \mu_i)(z_{si} - \mu_i)^T$ denote empirical estimates of the mean vector and the covariance matrix for the latent representations of the $i$-th arm of a $k$-CAE across the set $S$. Also, let $W_{ij} = \sum_{s \in S} (z_{si} - z_{sj})(z_{si} - z_{sj})^T$, $W = \sum_{i<j} W_{ij}$. We define the $k$-coupled batch-normalized autoencoding tuple ($k$-CBNAE), $\Phi = (\{(\mathcal{E}_i, \mathcal{D}_i, r_i)\}_{i \in K}, c, \lambda)$, as a $k$-CAE whose latent representations satisfy $\mu_i = 0$, and $\text{diag}(V_i) = \text{diag}(I)$, for any input set $X$.

**Proposition 2.** *If $c$ is the squared Euclidean norm and the diagonal values of $W$ are not all identical, latent representations of the $k$-CBNAE minimizing the loss in Eq. 1 with $C_{\text{coupling}} > 0$ satisfy $|z_{si}(m) - z_{si}(\bar{m})| < \epsilon$, for any $1 \le m, \bar{m} \le p$, $s \in S$, $1 \le i \le k$, $\epsilon > 0$.* (Proof in supp. material)

Thus, latent representations that do not collapse onto a single dimension do not have a stable training path in the sense that, under a continuous probability model for $z_{si}|z_{sj}$ (Section 2.4), such coupled representations are of measure zero.

### 2.3 Mini-batch singular value based normalization

Estimates of the covariance matrix are increasingly inaccurate for smaller batch sizes and larger latent dimensionalities. We propose an alternative that entails scaling the latent representation by the *narrowest* dimension. This can be formally evaluated as the smallest singular value of the batch matrix. $C_{\text{coupling}}$ can thus be written as:

$$C_{\text{coupling}} = \sum_{s \in S} \sum_{i<j} \frac{\|\bar{z}_{si} - \bar{z}_{sj}\|^2}{\min\left\{\sigma^2_{\min}(\bar{\mathbf{B}}_i), \sigma^2_{\min}(\bar{\mathbf{B}}_j)\right\}}, \tag{6}$$

where $\sigma_{\min}(\bar{\mathbf{B}}_i)$ is the smallest singular value of $\bar{\mathbf{B}}_i$, and $\bar{\mathbf{B}}_i$ is the $n \times p$ mini-batch matrix of the $i$-th autoencoder whose latent representation is batch normalized [11] (Eq. 3). We will refer to the coupling cost based on Eq. 6 as $C_{\text{MSV}}$, and that based on Eq. 4 as $C_{\text{FC}}$. Fig. 1B(iv) demonstrates that $C_{\text{MSV}}$ leads to representations with a well defined scale, that are qualitatively similar to those produced with the full covariance matrix based normalization for a 2D embedding. Importantly, $C_{\text{MSV}}$ is more robust against "the curse of dimensionality" compared to $C_{\text{FC}}$ (Fig. 1C-D). Moreover, the power iteration method offers an efficient algorithm to calculate the minimum singular value, sidestepping full eigendecomposition [19] (supp. material).

### 2.4 Probabilistic setting

While we pose our approach in a deterministic setting, here we show that the objective function in Eq. 5 is equivalent to the log-likelihood of a discriminative probabilistic model for i.i.d. observations:

$$\sum_{s \in S} \log p(x_{st}, x_{se}, \hat{z}_{st} | \hat{z}_{se}) = \sum_{s \in S} \log p(x_{st} | \hat{z}_{st}, \hat{z}_{se}) + \log p(\hat{z}_{st} | \hat{z}_{se}) + \log p(x_{se} | \hat{z}_{se})$$

$$= \sum_{s \in S} \log p(x_{st} | \hat{z}_{st}) + \log p(\hat{z}_{st} | \hat{z}_{se}) + \log p(x_{se} | \hat{z}_{se}), \tag{7}$$

where we assume that $x_{se}$ is independent of $x_{st}$ and $\hat{z}_{st}$ given $\hat{z}_{se}$, and $x_{st}$ is independent of $\hat{z}_{se}$ given $\hat{z}_{st}$. When $x_{st}$ denotes the $\log(\bullet + 1)$ transform of the transcriptomic readout for sample $s$ and $x_{se}$ denotes the sparse PC representation of the electrophysiology recordings for the same sample, we model the relevant conditional probabilities as $x_{st}|\hat{z}_{st} \sim \mathcal{N}(\widetilde{x}_{st}, \sigma_t^2 I)$, $x_{se}|\hat{z}_{se} \sim \mathcal{N}(\widetilde{x}_{se}, \sigma_e^2 I)$, and $\hat{z}_{st}|\hat{z}_{se} \sim \mathcal{N}(\hat{z}_{se}, \lambda^{-1} I)$. Then,

$$\sum_{s \in S} \log p(x_{st}, x_{se}, \hat{z}_{st} | \hat{z}_{se}) = \frac{-1}{2} \sum_{s \in S} \sigma_t^{-2} ||x_{st} - \widetilde{x}_{st}||_2^2 + \sigma_e^{-2} ||x_{se} - \widetilde{x}_{se}||_2^2 + \lambda ||\hat{z}_{st} - \hat{z}_{se}||_2^2 + \text{const}. \tag{8}$$

Therefore, maximizing the log-likelihood in Eq. 7 is equivalent to minimizing

$$\sum_{s \in S} ||x_{st} - \widetilde{x}_{st}||_2^2 + \alpha ||x_{se} - \widetilde{x}_{se}||_2^2 + \lambda ||\hat{z}_{st} - \hat{z}_{se}||_2^2, \tag{9}$$

which is equivalent to Eq. 5. Here, $\alpha = \sigma_t^2/\sigma_e^2$, and $\lambda$ is proportional to the precision in cross-modal latent variable estimation.

Note that the roles of the two modalities ($t$ and $e$) can be interchanged in Eq. 7. Moreover, Fig. 2B suggests that the individual cell types are well approximated by hyperellipsoids. Therefore, fitting a Gaussian mixture model to the encodings provides an efficient prior distribution for $p(\hat{z}_{se})$ (or $p(\hat{z}_{st})$), and produces a generative model for multi-modal datasets.

The cross-modal term in Eq. 9 is equivalent to the KL-divergence between two Gaussian distributions with identical diagonal covariances. Therefore, by removing the constraints on the latent space covariance matrices, we can obtain another generalization of Eq. 5 as $C_\Phi = C_{\text{recon}} + \lambda \sum_s D_{\text{KL}}(\hat{z}_{st}, \hat{z}_{se})$.

Lastly, while we used a Gaussian observation model with equal variances on the log-transformed transcriptomic data (a single output (mean) per gene), using non-identical variances as well as other distributions, such as the zero-inflated negative binomial model [12, 17, 20], is straightforward. In these cases, the decoding network would simply output parameters of the observation model for likelihood calculations (e.g., both mean and variance rather than just the mean).

## 3 Datasets

We used the MNIST dataset [21] to illustrate the effects of normalization strategies on the representations. We used a publicly available scRNA-seq dataset [22] (referred herein as the FACS dataset) to compare $C_{\text{FC}}$ with $C_{\text{MSV}}$ (Fig. 1C-D) and for experiments related to identifying shared and distinct cell types in multi-modal data (Fig. 5). Lastly, we used a novel dataset obtained with Patch-seq technology [23] to demonstrate the merit of our approach for the analysis of multi-modal datasets. This dataset consists of expression profiles of 1,252 genes (differentially expressed across established cell types, excluding sex/mitochondrial genes) across 2,945 neurons, and electrophysiological recordings of 4,637 neurons in mouse visual cortex. The electrophysiological recordings were obtained and summarized with a set of 54 sparse principle components (sPCA features) as obtained by Gouwens *et al.* [5]. 1,518 of these neurons were profiled with both data modalities, and assigned 80 distinct transcriptomic type labels following the hierarchical clustering scheme of [22]. While we train on all available data, we report cross-validation results based on the 44 types of neurons that were (i) profiled in both modalities, and (ii) that have at least 6 representatives in the training set. We refer to cells that were characterized with both modalities (only a single modality) as paired (unpaired) cells.

## 4 Results

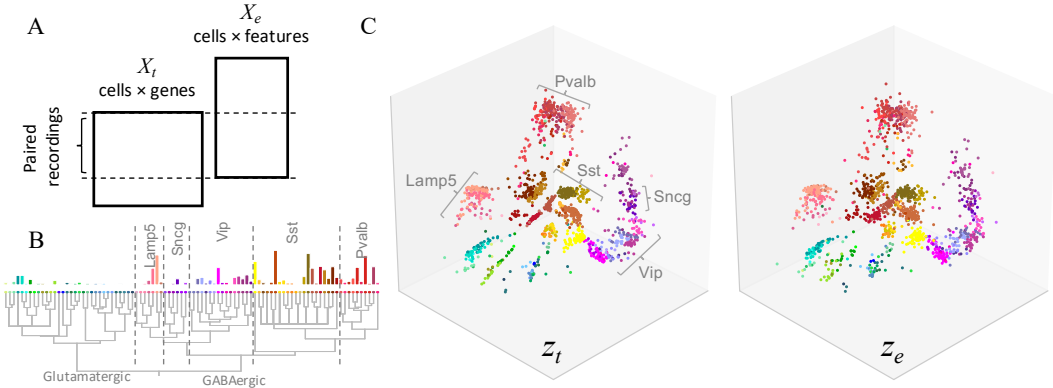

Figure 2: (**A**) 1,518 cells were profiled with both transcriptomic and electrophysiological modalities (paired recordings). (**B**) Relative distribution (bars) and hierarchical relationships (dendrogram) of ground truth cell type assignments (colors) for paired recordings, with well-known GABAergic cell classes annotated. (**C**) 3D coupled autoencoder based representations $z_t$ and $z_e$ ($\lambda = 1$) are qualitatively similar across the modalities.

We use multi-layer perceptrons to implement the encoder/decoder functions. Parameters of the resulting autoencoding architectures are fitted with stochastic mini-batch training and the Adam optimizer [24]. Transcriptomic measurements suffer from *gene dropout*, where the experiment fails to detect an expressed gene [25]. We use Dropout regularization [26] (i.e., Bernoulli noise) on the input layer as an augmentation strategy [27], which suggests a dropout probability of ~0.5. This agrees well with our experiments (Fig. S1). We set $p = 0.5$ for transcriptomic Dropout augmentation

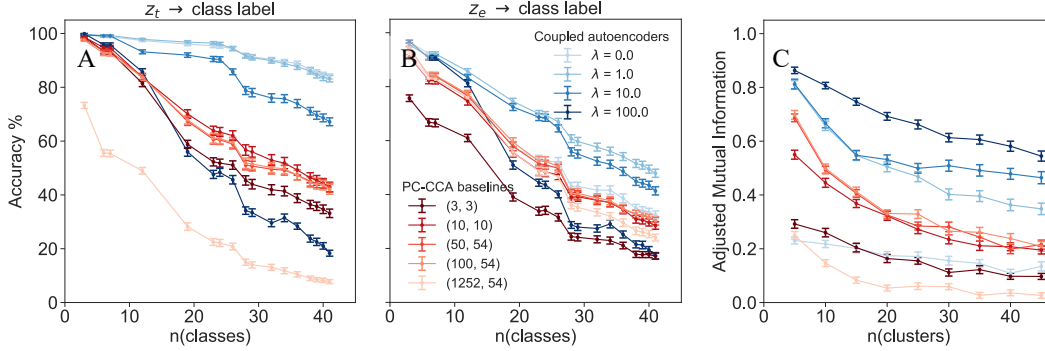

Figure 3: Cross-validated accuracy of quadratic classifiers trained on transcriptomic (**A**) and electrophysiology (**B**) representations in predicting transcriptomic cell classes at different resolutions of the hierarchy. (**C**) Adjusted mutual information for labels obtained with unsupervised clustering of the representations quantifies consistency between clusters across the modalities. 3D coupled autoencoder representations ($\lambda$=1,10) are more consistent with an established cell type hierarchy, allow for accurate cross-modal prediction of cell classes, and are more consistent across modalities compared to 3D CCA representations.

in all downstream analyses. In the same vein, we add i.i.d. Gaussian noise (and $p = 0.1$ Bernoulli noise) to the sPCA features of the electrophysiology measurements. See supp. material for additional details on the architecture and hyperparameters.

**Degenerate representations:** Experiments to evaluate coupling functions were performed by providing the same data as input to the different coupled autoencoder agents. Dropout and random initialization of network weights ensured that the representations produced by the different encoders are not identical. Tests with the MNIST dataset Fig. 1B(i-ii) illustrate problems with the representations obtained with commonly used coupling functions (Eq.2-3 and Prop.1-2). Normalization with the full covariance matrix, Eq.4 solves the issue of collapsing representations, Fig1B(iii). Using the mini-batch minimum singular value for normalization (Eq.6) achieves qualitatively similar representations, Fig. 1B (iv). Full covariance matrix estimates are expected to become unreliable as the latent space dimensionality grows and/or the mini-batch size becomes small compared to the latent space dimensionality. Tests with the FACS dataset Fig. 1C-D show that larger latent space dimensionality as well as smaller batch sizes lead to sub-par reconstruction performance for normalization with the full covariance matrix compared to that with the mini-batch minimum singular value.

**Cross-modal transcriptomic type prediction with QDA:** A question of biological significance is whether one can predict the transcriptomic type of a neuron based on only electrophysiological recordings. We performed 50-fold cross-validation to evaluate this ability using the Patch-seq dataset. Coupled autoencoders were used to obtain 3D representations, $z_t$ and $z_e$, for the transcriptomic and electrophysiology data respectively, with different values of the coupling strength $\lambda$. Ground truth class labels were obtained based on different depths of the reference hierarchical tree (Fig. 2B). We fixed $\alpha = 0.1$ for all Patch-seq experiments.

To test whether $z_t$ captures transcriptomic cell type definitions, we trained a quadratic classifier (QDA) to predict cell type labels based on $z_t$ and show prediction accuracy (mean $\pm$ SE, $n = 50$ cross-validation sets) in Fig. 3A. We find that the encoders produce clustered, unimodal representations consistent with the transcriptomic definition of the cell type hierarchy of Tasic *et al.* This suggests that a Gaussian mixture is a good model for the latent representations, as evidenced by $> 80\%$ accuracy over more than 40 types with a 3D latent space (Fig. 3A, $\lambda = 0$). As $\lambda$ is increased, the greater emphasis on minimizing mismatches with the electrophysiology representation leads to a slight degradation of transcriptomic type prediction. With $\lambda = 1, 10$, we were able to obtain highly consistent representations of multi-modal identity (Fig. 2C) as reflected by the high classification accuracy in Fig. 3A-B. We performed this analysis using 3D representations obtained with CCA [7, 28] that use transcriptomic and electrophysiological data reduced by PCA (PC-CCA, tuples indicate number of principal components of transcriptomic and electrophysiological data used for CCA). Transcriptomic and electrophysiological data were projected onto the top 3 CCA components, followed by a whitening transformation to ensure that the scale for the representations is the same. Red plots in Fig. 3A shows that 3D projections obtained in this manner offer a weak alternative to analyze multi-modal cell identity.

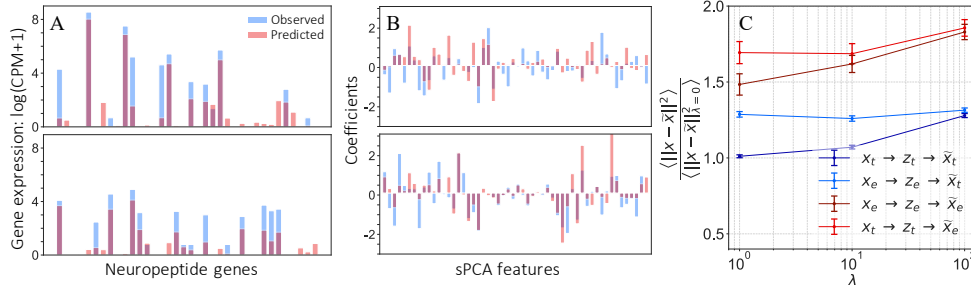

Figure 4: Cross-modal data prediction with 3D latent representations. Estimates of expression for a set of 37 peptidergic genes based on sPCA features (**A**), and of the sPCA features based on gene expression (**B**) for example test cells ($\lambda = 10$) show qualitative agreement of the predictions with the observations. (**C**) Quantifying $C_{\mathrm{recon}}$ with a reference of $\lambda = 0$ across the test set demonstrates the trade-off for $\lambda$ : increasing $\lambda$ makes the representations similar, leading to smaller differences between the same- (light colors) and cross-modal data (dark colors) prediction, and a higher $C_{\mathrm{recon}}$.

A similar analysis was performed using the electrophysiological representations, $z_e$, to test cross-modal prediction of transcriptomic types. Fig. 3B shows that the classifier performance is worse compared to Fig. 3A when $\lambda = 0$, which suggests that variations in the electrophysiology features do not completely overlap with variations in gene expression profiles. This is in line with the inconsistent clusters obtained in studies that consider single data modalities to define cell types. As $\lambda$ increases, $z_t$ and $z_e$ become more similar, and therefore allow cross modal prediction with better accuracy.

**Unsupervised cross modal type prediction:** We used unsupervised clustering to test the consistency of clusters obtained by coupled autoencoders to not be limited by the differential gene expression-based ground truth labels used for the supervised analysis. We fitted Gaussian mixture models with different component counts (E-M algorithm, 100 initializations) to the training data $z_t$ and $z_e$ independently, for each cross-validation set. Labels for $z_t$ and $z_e$ of the validation data were assigned based on their respective fitted mixture models. Fig. 3C shows the adjusted mutual information (mean $\pm$ SE, $n = 50$ cross-validation sets) as a measure of consistency of the labels obtained by such independent, unsupervised clustering of the representations. As $\lambda$ increases, the clusters become more consistent across modalities. The 3D CCA-based representations do not show distinct clusters, and consequently the consistency of labels unsupervised clustering is low overall.

**Analysis of reconstruction error as a function of** $\lambda$**:** The representations obtained by coupled autoencoders enable prediction of gene expression profiles from electrophysiological features and vice versa. Examples of such cross modal data predictions (Fig. 4A-B) based on very low dimensional ($d = 3$) representations capture salient features of the data already. To quantify the effect of imposing a penalty on representation mismatches when it comes to the cross modal data prediction task, we compared $C_{\mathrm{recon}}$ for data reconstructions based on coupled representations ($\lambda > 0$) to that obtained by setting $\lambda = 0$. Fig. 4C demonstrates that for the Patch-seq dataset, increasing $\lambda$ leads to worse reconstruction accuracy as expected. While the difference is small for predicting transcriptomic data, it is larger for electrophysiological feature prediction as a consequence of using $\alpha < 1$ (Section 2.4).

**Cell type discovery:** For partially paired datasets (Fig. 2A), an important problem is whether cell types not observed in some of the modalities can be uncovered by the alignment method. To test this, we split the FACS dataset into two subsets ($A$ and $B$), where samples of four cell types were allowed to be in only one of the two subsets. From among the cell types shared across $A$ and $B$, we considered 1/3 of the cells 'paired' based on (i) their cell type label, (ii) similarity of peptidergic gene expression [29], and (iii) distance in a representation obtained for the complete FACS dataset by a single autoencoder (see supp. methods for details). Fig. 5A shows the representations $z_A$ and $z_B$ obtained by the coupled autoencoder for the two subsets. Our results demonstrate that (i) types unique to subset $A$ appear in $z_A$ in positions that are not occupied by other cell types in $z_B$ and vice versa, whereas (ii) a type present in both subsets for which no cells were marked as paired occupied similar positions in $z_A$ and $z_B$. To quantify this observation, we calculated the nearest neighbor distance in $z_B$ for the types unique to subset $A$ by using their positions from $z_A$ (and vice versa), Fig. 4B. This simple quantification already shows that samples of types unique to subset $A$ can easily be distinguished from other types in subset $B$. This proof-of-principle experiment suggests that coupling representations in this manner can serve as a framework to discover shared and distinct cell types from aligned datasets, for data obtained from different modalities, brain regions, or species.

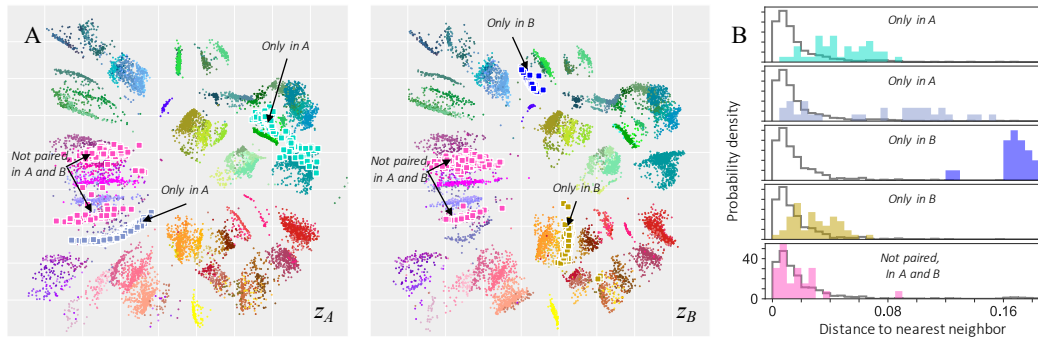

Figure 5: Coupled autoencoders can facilitate discovery of cell types unique to a single modality (**A**) 2D representations of two subsets created from the FACS dataset, with sparse (∼1/3) fraction of samples marked as paired. Colors: cell type annotations of [22]. Arrows: selected types exclusively placed in only one of the two subsets, or present in both subsets but with no samples considered as paired. The representations are qualitatively similar, with types unique to each subset appearing in distinct, non-overlapping locations. The type shared across the subsets but not considered as paired appears in similar positions. (**B**) Nearest-neighbor distance distributions for test cells ('paired' types are in the outlined distribution) in the 2D representation space supports these observations ($p < 0.01$ for top four rows, $p = 0.89$ for bottom row, 2-sample K-S test).

## 5 Discussion

We presented a method to identify the type of a cell based on observations from a single modality such that the identity would be consistent if the assignment was based on a different modality. While our method is applicable to cross-modal learning in general, our motivation stems from recent experimental developments in high-throughput, multi-modal profiling of neurons [30, 23]. In this study, we have demonstrated a surprising level of cross-modal predictive ability across transcriptomic and electrophysiological recordings. Specifically, we showed that the transcriptomic class can be predicted with ∼80% accuracy from electrophysiological recordings when the transcriptomic hierarchy is resolved into 15 classes, and with ∼70% accuracy when it is resolved into 25 classes ($\lambda = 10$ results). As datasets grow, we expect the performance to improve even in the absence of further technical development since many cell types in our dataset have a small number of samples.

While we focused on the correspondence problem between transcriptomics and electrophysiology($k = 2$), we presented the technical development of $k$-coupled autoencoders in full generality. Therefore, our method is applicable to the joint alignment of additional modalities.

The utility of autoencoders to obtain low dimensional representations of transcriptomic data, as well as the biological interpretation of such representations have been explored in recent works [17]. Here, we demonstrated the utility of the coupled autoencoder approach in obtaining such correspondence between modalities. We studied the potential pitfalls of coupling functions, and proposed a novel and practical function based on calculating the smallest singular value of the batch matrix.

We derived the distributions that establish an equivalence between our original deterministic approach and a discriminative probabilistic model. We also studied different generalizations of our objective function using this relationship. Finally, we proposed fitting a Gaussian mixture model to the latent representation *after* training, which provides an efficient generative model. Methodological improvements addressing potentially unshared variabilities across modalities, and joint, efficient learning of a generative model are promising avenues for future research.

Finally, we explored the ability of our method to identify cell types that are sampled only by a subset of characterization modalities. Such problems are frequently encountered due to sampling biases of the different experimental modalities and protocols used to characterize cells. We demonstrated that our method can (i) disambiguate types that may not be observed in all modalities, and (ii) obtain a coherent, well constrained embedding in the absence of pairing information for types that are sampled by multiple modalities (Fig. 5).

**Codes and Data:** Code repository: https://github.com/AllenInstitute/coupledAE. MNIST and FACS datasets are publicly available; Patch-seq dataset will be released by collaborators at a later date.

## Acknowledgements

We wish to thank the Allen Institute for Brain Science founder, Paul G Allen, for his vision, encouragement and support.

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
