[Supplementary Material]

# Supplementary Material

## Proofs for Propositions 1 and 2

**Definition.** *We define the k-coupled autoencoding tuple (k-CAE) $\Phi$ as*

$$\Phi = (\{(\mathcal{E}_i, \mathcal{D}_i, r_i)\}_{i \in K}, c, \lambda),$$

*where $K$ is an ordered, finite index set, $\mathcal{E}_i$, $\mathcal{D}_i$ are continuous operators such that they can express any linear transformation, $\operatorname{codomain}(\mathcal{E}_i) = \operatorname{domain}(\mathcal{D}_j)$, $i, j \in K$, $\lambda \geq 0$, and $r_i$ and $c$ are non-negative convex functions.*

**Definition.** *Let $X$ denote a set of inputs, $X = \{(x_{s1}, x_{s2}, \ldots, x_{sk}), s \in S\}$. The loss of the k-CAE $\Phi$ on $X$ is defined as*

$$C_\Phi(X) = C_{\operatorname{recon},\Phi}(X) + \lambda C_{\operatorname{coupling},\Phi}(X), \tag{1}$$

*where*

$$C_{\operatorname{recon},\Phi}(X) = \sum_{s \in S} \sum_{i \in K} r_i(x_{si} - \mathcal{D}_i(\mathcal{E}_i(x_{si}))),$$

$$C_{\operatorname{coupling},\Phi}(X) = \sum_{s \in S} \sum_{\substack{i,j \in K, \\ i < j}} c(\mathcal{E}_i(x_{si}) - \mathcal{E}_j(x_{sj})).$$

We assume $x_{si} \in \mathbb{R}^{p'}$, $z_{si} = \mathcal{E}_i(x_{si}) \in \mathbb{R}^p$, $p < p'$, and $|S| > 1$. In what follows, we simplify the notation for the coupling cost as $C_c^\Phi = C_{\operatorname{coupling},\Phi}(X)$.

**Remark.** *The reconstruction loss $C_{\operatorname{recon},\Phi}(X)$ is invariant under invertible transformations of the latent representations $z_{si} = \mathcal{E}_i(x_{si})$. In particular, for an invertible matrix $T$, $r_i(x_{si} - \mathcal{D}_i(\mathcal{E}_i(x_{si}))) = r_i(x_{si} - \mathcal{D}_i(z_{si})) = r_i(x_{si} - \widehat{\mathcal{D}}_i(\widehat{z}_{si}))$, where $\widehat{\mathcal{D}}_i = \mathcal{D}_i \circ T^{-1}$, and $\widehat{z}_{si} = T z_{si}$.*

We also note, for a diagonal matrix $A$ with non-negative diagonal entries, and a matrix $B$ with non-negative diagonal entries, we have

$$\operatorname{tr}(AB) = \sum_m A(m, m) B(m, m) \leq \max_m A(m, m) \operatorname{tr}(B). \tag{2}$$

**Proposition 1.** *Latent representations of the k-CAE minimizing the loss in Eq. 1 with $C_{\operatorname{coupling}} > 0$ satisfy $\|z_{si}\| < \epsilon$, for any norm $\|\cdot\|$, input set $X$, $\epsilon > 0$, and all $s, i$.*

*Proof.* Let $\Phi = (\{(\mathcal{E}_i, \mathcal{D}_i, r_i)\}_{i \in K}, c, \lambda)$ denote an optimal k-CAE on input set $X$ with latent representations $z_{si} = \mathcal{E}_i(x_{si})$, $\max_{s,i} \|z_{si}\| \geq \epsilon$, for some $\epsilon > 0$. Consider

$$\widehat{\Phi} = (\{(\beta I \circ \mathcal{E}_i, \mathcal{D}_i \circ \beta^{-1} I, r_i)\}_{i \in K}, c, \lambda) \tag{3}$$

$$\beta = \min\left(\frac{1}{2}, \frac{\epsilon}{2 \max_{s,i} \|z_{si}\|}\right). \tag{4}$$

The latent representations due to $\widehat{\Phi}$, $\widehat{z}_{si} = \beta z_{si}$, satisfy $\|\widehat{z}_{si}\| = \beta \|z_{si}\| < \epsilon$ and $\sum_{s \in S, i < j} c(\widehat{z}_{si} - \widehat{z}_{sj}) \leq \sum_{s \in S, i < j} \beta c(z_{si} - z_{sj}) < \sum_{s \in S, i < j} c(z_{si} - z_{sj})$. Thus, $C_{\operatorname{recon},\widehat{\Phi}}(X) = C_{\operatorname{recon},\Phi}(X)$ and $C_{\operatorname{coupling},\widehat{\Phi}}(X) < C_{\operatorname{coupling},\Phi}(X)$, contradicting optimality with $\max_{s,i} \|z_{si}\| \geq \epsilon$. $\quad\square$

Let $\mu_i = \frac{1}{|S|}\sum_{s\in S} z_{si}$, $V_i = \frac{1}{|S|-1}\sum_{s\in S}(z_{si} - \mu_i)(z_{si} - \mu_i)^T$ denote empirical estimates of the mean vector and the covariance matrix for the latent representations of the $i$-th agent of a $k$-CAE across the set $S$. Also, let $W_{ij} = \sum_{s\in S}(z_{si} - z_{sj})(z_{si} - z_{sj})^T$, $W = \sum_{i<j} W_{ij}$, and $\mathbf{1}$ denote a $p \times 1$ vector with $\mathbf{1}(m) = 1$ for all $m$.

**Definition.** *We define the $k$-coupled batch-normalized autoencoding tuple ($k$-CBNAE), $\Phi = (\{(\mathcal{E}_i, \mathcal{D}_i, r_i)\}_{i\in K}, c, \lambda)$, as a $k$-CAE whose latent representations satisfy $\mu_i = 0$, and $\mathrm{diag}(V_i) = \mathrm{diag}(I)$, for any input set $X$.*

**Proposition 2.** *If $c$ is the squared Euclidean norm and the diagonal values of $W$ are not all identical, latent representations of the $k$-CBNAE minimizing the loss in Eq. 1 with $C_{\text{coupling}} > 0$ satisfy $|z_{si}(m) - z_{si}(\bar{m})| < \epsilon$, for any $1 \le m, \bar{m} \le p$, $s \in S$, $1 \le i \le k$, $\epsilon > 0$.*

*Proof.* Let $\Phi = (\{(\mathcal{E}_i, \mathcal{D}_i, r_i)\}_{i\in K}, c, \lambda)$ denote an optimal $k$-CBNAE on input set $X$ with latent representations $z_{si} = \mathcal{E}_i(x_{si})$, satisfying $|z_{si}(m) - z_{si}(\bar{m})| \ge \epsilon$ for some $1 \le m, \bar{m} \le p$, $s \in S$, $1 \le i \le k$, $\epsilon > 0$. Let $1 \le n \le p$ be such that $b^T W b$ is minimized for the $p \times 1$ vector $b$ with $b(m) = 0$ for $m \ne n$ and $b(n) = 1$: $b^T W b \le \gamma W(m, m)$ for any $m$, $\gamma \le 1$. Note that

$$
\begin{aligned}
C_c^\Phi &= \mathrm{tr}(W), & (5)\\
b^T W b &< \mathrm{tr}(W)/p, & (6)\\
\mathrm{tr}(\mathbf{1}b^T W) &= \sum_m W(n, m) \\
&\le \sum_m \sqrt{W(n,n)W(m,m)} < \sum_m \frac{W(n,n) + W(m,m)}{2} < \mathrm{tr}(W), & (7)
\end{aligned}
$$

where the strict inequality in Eq. 6 is due to the non-identical diagonal values assumption, the first inequality in Eq. 7 is due to the Cauchy-Schwartz inequality, and the strict inequalities in Eq. 7 are due to inequality of arithmetic and geometric means and the non-identical diagonal values assumption.

$|V_i(n, m)| \le 1$ due to the Cauchy-Schwartz inequality. Moreover,

$$
\min_m V_i(n, m) < 1 \tag{8}
$$

because equality implies $|z_{si}(m) - z_{si}(\bar{m})| < \epsilon$, for any $1 \le m, \bar{m} \le p$, $s \in S$, $1 \le i \le k$, $\epsilon > 0$, again due to Cauchy-Schwartz.

Consider $\widehat{\Phi} = (\{(M_i \circ \mathcal{E}_i, \mathcal{D}_i \circ M_i^{-1}, r_i)\}_{i\in K}, c, \lambda)$, $0 < \beta < 1$, where

$$
M_i = D_i \mathbf{1}b^T + \beta D_i, \tag{9}
$$

and $D_i$ is a diagonal matrix with diagonal entries

$$
D_i(m, m) = [1 + \beta^2 + 2\beta V_i(n, m)]^{-1/2}. \tag{10}
$$

Therefore, $\frac{1}{1+\beta} \le D_i(m, m) \le \frac{1}{1-\beta}$, $(D_i(m,m) - D_j(m,m))^2 \le \frac{4\beta^2}{(1-\beta^2)^2}$. By Eq. 8,

$$
\mathrm{tr}(D_i) < \frac{p}{1-\beta} \tag{11}
$$

$$
\mathrm{tr}(D_i D_i) < \frac{p}{(1-\beta)^2}. \tag{12}
$$

Then,

$$
\widehat{m}_i = \frac{1}{|S|}\sum_{s\in S}\widehat{z}_{si} = (D_i \mathbf{1}b^T + \beta D_i)\frac{1}{|S|}\sum_{s\in S} z_{si} = 0 \tag{13}
$$

$$
\begin{aligned}
\widehat{V}_i &= \frac{1}{|S|-1}\sum_{s\in S}(\widehat{z}_{si} - \widehat{m}_i)(\widehat{z}_{si} - \widehat{m}_i)^T = M_i\left(\frac{1}{|S|-1}\sum_{s\in S} z_{si}z_{si}^T\right)M_i^T \\
&= D_i[\mathbf{1}b^T V_i b \mathbf{1}^T + \beta(\mathbf{1}b^T V_i + V_i b \mathbf{1}^T) + \beta^2 V_i]D_i^T. \tag{14}
\end{aligned}
$$

Using Eq. 10, the diagonal entries of $\widehat{V}_i$ are

$$
\widehat{V}_i(m, m) = D_i(m, m)^2 [V_i(n, n) + 2\beta V_i(n, m) + \beta^2 V_i(m, m)] = 1. \tag{15}
$$

We now calculate upper bounds for two intermediate quantities, $\Gamma_1$ and $\Gamma_2$:

$$\Gamma_1 = \sum_{s \in S} \sum_{i<j} [M_i(z_{si} - z_{sj})]^T [M_i(z_{si} - z_{sj})] \tag{16}$$

$$= \sum_{i<j} \mathrm{tr}([D_i \mathbf{1} b^T + \beta D_i] W_{ij} [b \mathbf{1}^T D_i^T + \beta D_i^T]) \tag{17}$$

$$= \sum_{i<j} \mathrm{tr}(\mathbf{1}^T D_i^T D_i \mathbf{1}(b^T W_{ij} b)) + 2\beta \sum_{i<j} \mathrm{tr}(D_i^T D_i \mathbf{1} b^T W_{ij}) + \beta^2 \sum_{i<j} \mathrm{tr}(D_i^T D_i W_{ij}) \tag{18}$$

$$= \sum_{m} D_i(m,m)^2 \, \mathrm{tr}(b^T W b) + 2\beta \sum_{i<j} \mathrm{tr}(D_i^T D_i \mathbf{1} b^T W_{ij}) + \beta^2 \sum_{i<j} \mathrm{tr}(D_i^T D_i W_{ij}) \tag{19}$$

$$< \frac{p b^T W b}{(1-\beta)^2} + \frac{2\beta}{(1-\beta)^2} \, \mathrm{tr}(\mathbf{1} b^T W) + \frac{\beta^2}{(1-\beta)^2} \, \mathrm{tr}(W) \tag{20}$$

$$< \frac{\gamma C_{\mathrm{c}}^{\Phi}}{(1-\beta)^2} + \frac{2\beta C_{\mathrm{c}}^{\Phi}}{(1-\beta)^2} + \frac{\beta^2 C_{\mathrm{c}}^{\Phi}}{(1-\beta)^2} \tag{21}$$

$$< \frac{(\gamma + 3\beta) C_{\mathrm{c}}^{\Phi}}{(1-\beta)^2}, \tag{22}$$

where we used Eq. 2 in Eq. 20, and Eqs. 5, 6 and 7 in Eq. 22.

$$\Gamma_2 = \sum_{s \in S} \sum_{i<j} [(M_i - M_j) z_{sj}]^T [(M_i - M_j) z_{sj}] \tag{23}$$

$$= \sum_{i<j} \mathrm{tr}\left( (M_i - M_j) \sum_{s \in S} z_{sj} z_{sj}^T (M_i - M_j)^T \right) \tag{24}$$

$$= (|S| - 1) \sum_{i<j} \mathrm{tr}((M_i - M_j) V_j (M_i - M_j)^T) \tag{25}$$

$$= (|S| - 1) \sum_{i<j} \mathrm{tr}((D_i - D_j)^T (D_i - D_j)(\mathbf{1} b^T + \beta I) V_j (b \mathbf{1}^T + \beta I)) \tag{26}$$

$$\leq \frac{4\beta^2 (|S| - 1)}{(1 - \beta^2)^2} \sum_{i<j} \mathrm{tr}((\mathbf{1} b^T + \beta I) V_j (b \mathbf{1}^T + \beta I)) \tag{27}$$

$$\leq \frac{4\beta^2 (|S| - 1)}{(1 - \beta^2)^2} \sum_{i<j} \left[ \mathrm{tr}(b^T V_j b \mathbf{1}^T \mathbf{1}) + 2\beta \, \mathrm{tr}(\mathbf{1} b^T V_j) + \beta^2 \, \mathrm{tr}(V_j) \right] \tag{28}$$

$$< \frac{4p\beta^2 (|S| - 1)}{(1 - \beta^2)^2} \sum_{i<j} \left[ b^T V_j b + 2\beta + \beta^2 \right] \tag{29}$$

$$< \frac{4p\beta^2 (|S| - 1)}{(1 - \beta^2)^2} \frac{p(p-1)}{2} (1 + 2\beta + \beta^2) < \frac{2p^3 \beta^2 (|S| - 1)(1 + 3\beta)}{(1 - \beta^2)^2} \tag{30}$$

$$< \frac{f \beta^2}{(1 - \beta)^2}, \tag{31}$$

where $f = 8p^3 (|S| - 1) > 0$, $0 < \beta < 1$, and we used Eq. 2 in Eq. 27, and Eq. 8 in Eq. 29.

$$C_{\mathrm{c}}^{\widehat{\Phi}} = \sum_{s \in S} \sum_{i<j} (M_i z_{si} - M_j z_{sj})^T (M_i z_{si} - M_j z_{sj}) \tag{32}$$

$$= \sum_{s \in S} \sum_{i<j} [M_i(z_{si} - z_{sj}) + (M_i - M_j) z_{sj}]^T [M_i(z_{si} - z_{sj}) + (M_i - M_j) z_{sj}] \tag{33}$$

$$\leq \Gamma_1 + \Gamma_2 + 2\sqrt{\Gamma_1 \Gamma_2} \tag{34}$$

$$< \frac{(\gamma + 3\beta) C_{\mathrm{c}}^{\Phi}}{(1-\beta)^2} + \frac{f \beta^2}{(1-\beta)^2} + \frac{2\sqrt{(\gamma + 3\beta) f \beta^2 C_{\mathrm{c}}^{\Phi}}}{(1-\beta)^2} \tag{35}$$

$$< \frac{(\gamma + 3\beta) C_{\mathrm{c}}^{\Phi} + f\beta + 4\beta \sqrt{f C_{\mathrm{c}}^{\Phi}}}{(1-\beta)^2}, \tag{36}$$

where the inequality in Eq. 34 is due to Cauchy-Schwartz. Then,

$$C_c^\Phi - C_c^{\widehat{\Phi}} \geq \frac{((1-\beta)^2 - \gamma - 3\beta)C_c^\Phi - f\beta - 4\beta\sqrt{fC_c^\Phi}}{(1-\beta)^2} \tag{37}$$

$$\geq \frac{C_c^\Phi \beta^2 - [5C_c^\Phi + f + 4\sqrt{fC_c^\Phi}]\beta + C_c^\Phi(1-\gamma)}{(1-\beta)^2}. \tag{38}$$

The determinant of the quadratic (in $\beta$) $Q = C_c^\Phi \beta^2 - [5C_c^\Phi + f + 4\sqrt{eC_c^\Phi}]\beta + C_c^\Phi(1-\gamma)$ is positive, the coefficient of the quadratic term, $C_c^\Phi$, is positive, and the coefficient of the linear term, $-(5C_c^\Phi + f + 4\sqrt{fC_c^\Phi})$, is negative. When $\beta = 0$, $Q > 0$. Therefore, both roots of $Q$, $r_1$ and $r_2$, are positive ($0 < r_1 \leq r_2$), and for $\beta < \min(\frac{1}{2}, \frac{r_1}{2})$,

$$C_c^{\widehat{\Phi}} < C_c^\Phi. \tag{39}$$

Finally, the latent representations due to $\widehat{\Phi}$, $\widehat{z}_{si} = M_i z_{si}$, satisfy

$$|\widehat{z}_{si}(m) - \widehat{z}_{si}(\bar{m})| = |(D_i(m,m) - D_i(\bar{m},\bar{m}))z_{si}(n)$$
$$+\beta(D_i(m,m)z_{si}(m) - D_i(\bar{m},\bar{m})z_{si}(\bar{m}))| \tag{40}$$

$$\leq \frac{2\beta}{1-\beta^2}\|z_{si}\| + \frac{\beta}{1-\beta}\|z_{si}\| + \frac{\beta}{1-\beta}\|z_{si}\| \tag{41}$$

$$\leq \frac{4\beta}{1-\beta}\|z_{si}\| \tag{42}$$

so that $|\widehat{z}_{si}(m) - \widehat{z}_{si}(\bar{m})| < \epsilon$ for $\beta \leq \frac{\epsilon}{4\max_{s,i}\|z_{si}\|}$.

Thus, for

$$\beta = \frac{1}{2}\min(\frac{1}{2}, \frac{r_1}{2}, \frac{\epsilon}{4\max_{s,i}\|z_{si}\|}), \tag{43}$$

$C_{\text{recon},\widehat{\Phi}} = C_{\text{recon},\Phi}$ and $C_{\text{coupling},\widehat{\Phi}} < C_{\text{coupling},\Phi}$, contradicting optimality with $|\widehat{z}_{si}(m) - \widehat{z}_{si}(\bar{m})| \geq \epsilon$ for some $1 \leq m, \bar{m} \leq p$, $s \in S$, $1 \leq i \leq k$, any $\epsilon > 0$. $\qquad\square$

## Calculation of the minimum singular value

We are interested in calculating the minimum singular value of a *full-rank* matrix $\mathbf{B}$. Noting that the singular values of $\mathbf{B}$ are square roots of the eigenvalues of $\mathbf{B}^T\mathbf{B}$, it suffices to calculate the minimum eigenvalue of $\mathbf{A} = \mathbf{B}^T\mathbf{B}$. The following basic power iteration algorithm obtains an estimate: We

---

**Algorithm 1** Simple power iteration to find minimum eigenvalue, $\varepsilon$ determines the precision of convergence.

---

initialize $x_0$ randomly, and $i \leftarrow 0$
solve for $x_1$: $\mathbf{A}x_1 = x_0$
**while** $|x_{i+1} - x_i| > \varepsilon$ **do**
    solve for $x_{i+1}$: $\mathbf{A}x_{i+1} = x_i$
    $i \leftarrow i + 1$
**end while**
return $x_{i+1}$

---

refer the reader to the vast literature on power iterations (e.g. [Baglama *et al.*, 17]) for more efficient implementations. Note that the algorithm corresponds to a power iteration of $\mathbf{A}^{-1}$ without explicit matrix inversion. Therefore, the result will be an approximation to the largest eigenvalue of $\mathbf{A}^{-1}$, which is the smallest eigenvalue of $\mathbf{A}$.

## Datasets, network architectures, and training

**Experiments with the MNIST dataset:** The architecture for the autoencoders processing MNIST transcriptomic data is Input($28\times28$) $\to$ Conv2D(10) $\to$ MaxPooling($2\times$) $\to$ Flatten() $\to$ Dropout(0.4)

$\rightarrow$ Dense(49) $\rightarrow$ Dense(49) $\rightarrow$ Dense(49) $\rightarrow$ Dense(2) $\rightarrow$ BatchNormalization(2D latent representation $z$) $\rightarrow$ Dense(49) $\rightarrow$ Dense(49) $\rightarrow$ Dense(196) $\rightarrow$ Reshape() $\rightarrow$ UpSampling(2$\times$) $\rightarrow$ Conv2D(10) $\rightarrow$ Conv2D(1) $\rightarrow$. pixels. Numbers in brackets indicate the number of units in the 2D convolutional (Conv2D) and fully connected (Dense) layers. All convolutional filters are of size (3$\times$3) pixels. The downsampling (MaxPool) and upsampling operations change the dimensions of the input to those layers by a factor of 2. The networks were trained for 500 epochs, with a batch size of

**Experiments with the Patch-seq dataset:** The architecture for the autoencoder processing the Patch-seq transcriptomic data is Input(1252) $\rightarrow$ Dropout($p$) $\rightarrow$ Dense(60) $\rightarrow$ Dense(60) $\rightarrow$ Dense(60) $\rightarrow$ Dense(60) $\rightarrow$ Dense(d) $\rightarrow$ Batch Normalization (latent representation $z_t$) $\rightarrow$ Dense(60) $\rightarrow$ Dense(60) $\rightarrow$ Dense(60) $\rightarrow$ Dense(60) $\rightarrow$ Dense(1252). For the electrophysiology data, the architecture is Input(54) $\rightarrow$ Dropout($p_e$) $\rightarrow$ Gaussian Noise($\sigma$) $\rightarrow$ Dense(50) $\rightarrow$ Dense(50) $\rightarrow$ Dense(50) $\rightarrow$ Dense(50) $\rightarrow$ Dense(d) $\rightarrow$ Batch Normalization (latent representation $z_e$) $\rightarrow$ Dense(50) $\rightarrow$ Dense(50) $\rightarrow$ Dense(50) $\rightarrow$ Dense(50) $\rightarrow$ Dense(54). Numbers in parentheses denote the number of units in that layer, the numbers of input/output units in each network match the number of input genes and electrophysiology features respectively.

Dropout was used to prevent overfitting for both autoencoder sub-networks. The Dense layers use the rectified linear function as the nonlinear transformation except for Dense($d$) layers and the last Dense layer for the electrophysiology data which do not use a non-linear transformation. Each mini-batch consists of randomly chosen samples of 75 paired cells and 25 unpaired cells. The architectures of the autoencoder networks for transcriptomics and electrophysiology were only roughly tuned, independently of one another to prevent overfitting and obtain interpretable representations. Tests were performed with latent dimensionality $d = 2$ and 3, and we analyze the results for $d = 3$. We set dropout probability $p = 0.5$ because the reconstruction error was found to be the least with this value. While we set $p = 0.1$ and $\sigma = 0.05$, we did not observe a significant change in performance within a large range for these parameters. We trained the coupled autoencoders with $\lambda \in \{0.0, 1.0, 10.0, 100.0\}$ and $\alpha = 0.1$, for which the results are presented in the main text. Training these networks for 2000 epochs with 500 gradient steps per epoch took $\sim$3 hours on a standard laptop with the 2.3GHz quad-core Intel Core i7 processor.

Figure S1: For the Patch-seq transcriptomic data, dropout probability of $p = 0.5$ results in the lowest $C_{\mathrm{recon}}$ on the validation set (mean $\pm$ SE, $n = 10$).

Figure S2: Ground truth cell type labels, and their hierarchical relationships based on Tasic *et al.*. Bar plot shows relative distribution of labels assigned to the 1,518 paired recordings in the Patch-seq data (same as Fig. 2B). The color code is chosen to reflect the hierarchical relationships, and serves as a visual guide to interpret the representations.

**Experiments with the FACS dataset:** The FACS dataset was used to compare coupling functions $C_{\mathrm{FC}}$ and $C_{\mathrm{MSV}}$ with different latent dimensionality and batch sizes. It was also used to evaluate whether coupled autoencoders can be used to discover cell types that are either present in only a subset of the datasets, or to uncover *a priori* unknown similarities between types across datasets. The network architecture for individual autoencoder agents in all of these experiments was similar to that used for the Patch-seq transcriptomic experiments, with the number of inputs set to 5,000 and number of units in the intermediate layers set to 100.

To test the utility of coupled autoencoders for cell type discovery, we first discarded non-neuronal cells and two outlier cell types, Meis2 Adamts19 and CR Lhx5 (see Tasic et al.) from the FACS dataset. The dataset was thereafter split into subset $A$ and subset $B$. Cells labeled as *Vip Igfbp6 Pltp* and *L5 IT VISp Hsd11b1 Endou* were present only in subset $A$, whereas those labeled as *Sst Hpse Sema3c* and *L5 PT VISp Chrna6* were present only in subset $B$. All other types were present in roughly equal proportion in the two subsets. We selected the top 5,000 genes based on their maximum expression values calculated over all the cells. The counts per million (CPM) normalized gene expression values were incremented by 1, and log transformed thereafter to use as input to the autoencoders.

To determine the cells that should be considered as paired across subsets $A$ and $B$, we first obtained a 2D representation for the complete FACS dataset with a single autoencoder. The resulting representation was used to obtain the Euclidean pairwise distance matrix $D_{AB}$ for all cells in subset $A$ from all cells in subset $B$. Next, considering only one cell type label at a time, we found the best pairing of cells across subsets $A$ and $B$ with the Hungarian algorithm using $-D_{AB}$ as the assignment cost. The pairings were further pruned using a cell type specific threshold imposed on the cross correlation coefficient of peptidergic gene expression values. All pairs for the cell type *Vip Chat Htr1f* were discarded. This process resulted in 4,397 paired samples among the 11,093 cells in subset $A$ and 11,272 in subset $B$. 10% of cells from types that were specific to either subset $A$ or $B$, or had no paired samples across the subsets were held back to serve as the test set.

Figure S3: 3D representations of the Patch-seq transcriptomic and electrophysiological datasets in the uncoupled setting ($\lambda = 0$) and coupled setting ($\lambda = 1.0, 10.0$). Colors indicate ground truth cell type labels

We used a warm start for the coupled networks during training. A single autoencoder was first trained for 5,000 epochs with 20 steps per epoch using only samples from the $4,397 \times 2$ unique cells paired across the subsets. The weights of this autoencoder were used to initialize both autoencoders in the coupled configuration. For tests with the coupled autoencoders, we used $\lambda = 1.0$ and $\alpha = 1.0$. Mini-batches consisted of 500 samples from each subset, out of which 200 were from among the paired set of cells. Training the coupled network for 10,000 epochs with 20 steps per epoch took 8 hours on a standard laptop with the 2.3GHz quad-core Intel Core i7 processor.