[Reviews · NeurIPS 2019]

Reviewer 1



The authors present novel work regarding the ability of coupled autoencoders to analyse multi-modal profiles of neurons. Previous work is well cited and the work is technically sound with well supported and clear claims. This work is highly significant as neuroscience labs around the world grapple with defining neuronal cell types; the problem has become especially acute given the explosion of high-throughput ‘big data’ transcriptomic datasets. Minor issues: Fig 4 legend typo “ephysiological” Line 252 missing word “one” between “only” and “of” here: “were allowed to be in only of the two subsets”

Reviewer 2



This is a very interesting paper on a highly pertinent question: how do neural cell type definitions align across different data modalities. The figures all still look a little rough and need to be brushed up for publication. l.80 E and D are limited to 'linear' transformations? Why is this? And how does that fitted to the, presumably, nonlinear MLPs used later? The notation in section 2.1. is unnecessarily cluttered, please shorten and clean up. For instance, alphas can be absorped in the lambdas? Please state clearly the difference to and novelty over citation [9]. Proposition seems trivial and unnecessary, it would seem fine to just say in plain text that this may happen. Moreover, I am rather confused by your complicated approach to the shrinkage problem as a trivial solution to the euclidean distances between hidden states. Would it not suffice to simply look at the L2 distances after normalization (i.e. dividing by the norm) or look at a normalized distance metric like cosine similarity? That would seem a much simpler solution than sections 2.2 and 2.3? The crossmodal prediction tasks seem very interesting but I am not sure if I understood the details, could you explain the results a bit more? (e.g. what predictors are used, what is predicted?) I am also a little uneasy about the clustering performed on top of the latent representations. How much is the latent geometry determined by the workaround (2.2.)? Is it surprising that is so regular and missing data fills in well? In unimodal cell type clustering there is a lot of uncertainty about the clusters and whether they are biologically meaningful. In this second stage clustering on the aligned latents it seems even more removed and harder to interpret whether this is actually something that falls in line with the underlying biology. Theoretically, should not the cell type identites be the ultimate latent (causal) factors that yield perfect alignment across data modalities – modulo intra-type variability?

Reviewer 3



Originality: Multimodal data is increasingly becoming available in various omics field. Notably in neuroscience, patch-seq has been recently developed to profile neurons both transcriptomically and electrophysiologically (Cadwell et al, 2016, Fuzik et al 2016). Now, the first large data sets are becoming available, yet analysis methods that can fully leverage the multimodal data sets are still largely missing (see Tripathy et al, 2018; Tripathy et al. 2017, Kobak et al. 2018). The present submission extends prior work in coupled autoencoder architecture to patch-seq and equips them with a new loss function for the coupling loss that does not allow for degenerate solutions. Quality: Overall the paper appears to be well done – it almost contains a bit too much material for such a short format. There are some aspects, however, which lessen my enthusiasm: Fig 2: I would *really* like to see the embedding produced by uncoupled autoencoders (lambda=0). Currently it's not clear if the coupled representation is driven primarily by transcriptomics, primarily by electrophysiology, etc. Fig. 3: It seems the representation by the transcriptome autoencoder is much better than that of the ephys encoder and titrating lambda effectively “contaminates” the good transcriptome representation with the poor ephys representation, judiging from the accuracy with different lambdas. While lambda=10 provides a good compromise, ideally one would like to enhance the worse of the two representations, not make both poorer. Fig 3: I could not find any details of the CCA implementation. Vanilla CCA requires inverting covariance matrices but with n=1518 and p=1252, covariance matrix in the transcriptomic space is very badly determined, likely resulting in strong overfitting. One would need to use regularized CCA (with cross-validation to select the ridge penalty) or at least strongly reduce the dimensionality with PCA before running CCA. Otherwise the comparison does not make a lot of sense for a trivial reason. The very poor CCA performance in panel C (see also line 238) suggests that it could have been the case. Fig. 3: I did not understand the results in C – in the supervised setting lambda = 0/10 lead to very good results in terms of transcriptomic representation (and ephys as well), but in C it by far leads to the worst, while lambda =100 (which is terrible in A/B) leads to the best. Please explain. -> The author reply has adressed the two last points in a convincing manner, so I adjusted my score. Clarity: lines 37-42: this is phrased as if coupled autoencoders do not "involve calculation of explicit transformation matrices and possibly parameters of multi-layer perceptrons" but in reality they do. Consider rephrasing. It is not clear from the text if Feng et al. used batch normalization. From a cursory look at Feng et al., they did not use it. According to this paper, this cannot yield meaningful results. Please explain. Lines 119-125: I could not understand what is claimed here. Proposition 2 and especially the sentence underneath look as if they are saying that k-CBNAE has some bad properties ("not have a stable training path", "measure zero", inequality with <\epsilon). But the authors probably want to argue that k-CBNAE has good properties. Lines 140 and following: The transition from the deterministic to the probabilistic setting was to quick for me and unmotivated, especially because the authors use this form by optimizing lambda later on. Please expand. lines 143-144: transcriptomic data and sparse PCA has not yet been described: the data are first mentioned in the next section line 166: if Fig 1C-D use FACS dataset and not MNIST dataset, you should say so in the caption of Fig 1. Fig 2: consider using 2D bottleneck for this figure instead of 3D. It's fine to use 3D for later analysis like in Fig. 3 if the authors prefer, but it seems much more natural to use 2D for scatter plot visualisation. line 168: why 1252 genes? Is it the number of "most variable" genes that were retained for this analysis? How was this done? line 171: 1518 neurons had both modalities, but 2945 neurons were sequenced. What about the remaining 2945-1518 neurons: if they were sequenced using Patch-seq, how come they do not have electrophysiological recordings? Lines 208-222: reorganize the text, it is hard to follow line 228 and Fig 3B: the text says that the performance of z_e -> class label increases with lambda and it would indeed make sense, however the figure shows that the perfomance drops when lambda grows above 1. Why? line 248: "using alpha<1 (section 2.4)" -- I did not understand this. What is this alpha, why was it <1 in the ephys space? Significance: Potentially high significance, but paper too premature.

[Author Response · NeurIPS 2019]

All reviewers agree that this contribution is timely, and recognize its potential impact. We thank **R1** in particular for providing an excellent, concise summary of our contributions. While no major technical/scientific issues were raised in any of the reviews, our manuscript is nevertheless stronger by virtue of incorporating clarifications requested by **R2**, **R3**, and the addition of relevant references and stronger baselines (see Fig. below) suggested by **R3**. To reiterate our central contributions: we provided a rigorous analysis and proposed a solution for a critical technical issue that now enables learning of interpretable representations with coupled autoencoders. We applied this development to an unprecedented patch-seq dataset consisting of thousands of samples. Our optimization framework provides a novel, principled way of assessing the cell type hypothesis (e.g. Zeng and Sanes, 2017), and the results suggest that neuronal identities can be consistent to a surprisingly high degree across transcriptomic(T) and electrophysiological(E) modalities.

Recognizing that the reviewers did not point to any technical or scientific flaws in the *Improvements* section, we respectfully hope that the clarifications and analysis provided here will warrant substantially higher scores.

**Misc. (R3)** The dataset consists of 1252 differentially expressed genes, selected after excluding sex/mitochondrial genes. E recordings on the 2945-1518 neurons did not satisfy predefined quality control criteria. Indeed, an important strength of our approach is the ability to work with partially matched datasets. **(R2,R3)** We have added these and other clarifications, annotated well-known classes and hierarchical structure (Exc vs. Inh, Sst, Vip, PValb classes etc. in Fig. 2C) to make the connection to biology evident, and included 2D and 3D $\lambda = 0$ representations for completeness.

**Sec. 2.1-2.4**: **(R3)** lines 37-42: There are no explicit transformation matrices required to go from one representation space to another for the coupled autoencoder. We rephrase this now to avoid confusion. **(R3)** line 248: $\alpha$ is first defined in Sec. 2.1, and used consistently in Sec. 2.2 and 2.4. As studied in Sec. 2.4, it represents the relative noise level in the different modalities. Since this ratio is not measured explicitly, we heuristically set $\alpha = 0.1$ for all patch-seq experiments to capture the understanding that the T data is of higher resolution and quality. **(R2)** line 80: $\mathcal{E}$ and $\mathcal{D}$ can indeed be nonlinear; the statement only implies that they are *at least* capable enough to represent any linear transformation. **(R2)** The objective function contains two or more reconstruction error terms, and so all $\alpha_i$'s cannot be absorbed into $\lambda$. **(R2)** The suggestion to use a distance metric following normalization of individual representations would not prevent representations from collapsing. Proposition 2 proves why such strategies are guaranteed to fail, and formalizes exactly this non-trivial understanding of the problem.

**Feng *et al*. 2014**: **(R2,R3)** Feng *et al.* do not specify any normalization (Batch Norm. (BN) paper appeared in 2015). tSNE transforms used in that paper hide the shrinking problem, and their representations display poor alignment for all parameter values (Fig. 11 in Feng *et al.* - squares vs. pluses). Without normalization, the representations asymptotically collapse to a point (Prop 1). With uncoordinated normalization (e.g., BN), they asymptotically collapse to a line (Prop 2, k-CBNAE). Our proposed solution ($C_{\mathrm{MSV}}$) avoids both problems, and is efficient and robust (Fig 1C,D and Sec. 2.3).

**Representation quality**: **(R2,R3)** Our optimization framework trades off the consistency of representations across modalities against the fidelity of representations to raw data. The ultimate test of whether coupled representations are biased by either modality is the cross modal data prediction ability (as quantified in Fig. 4C). Representations $z_t$ and $z_e$ in Fig. 2C show consistency across modalities (dot positions), and capture biologically relevant transcriptomic hierarchy of cell classes (colors, Fig 2B-C). **(R3)** line 228, Fig. 3B: As coupling ($\lambda$) increases, $z_t$ and $z_e$ become more consistent (Fig. 3C), at the expense of $z_t$ capturing less of the transcriptomic hierarchy.(Fig. 3A). It is precisely because of this 'handshake' that $\lambda = 10$ is marginally lower than $\lambda = 1$ in Fig. 3B. While we do not tune $\alpha$ and $\lambda$, as multimodal datasets mature, it would be appropriate to optimize these parameters based on cross-modal data prediction ability (e.g. $x_t \to z_t \to \tilde{x}_e$: start from raw T data $x_t$, obtain the representation $z_t$, and pass it through the E decoder to predict raw E data $x_e$). We explore this systematically in Fig. 4C, where we show within and across-modality data prediction accuracies relative to reconstruction accuracy of individual, uncoupled ($\lambda = 0$) networks. From among the coupling strengths evaluated, $\lambda = 10$ strikes a desirable balance between measures of consistency in the latent space (Fig. 3B,C, example in Fig. 2C) while capturing known cell type hierarchies (Fig. 3A), and prediction accuracy (Fig. 4).

**Revised Fig. 3. Coupled AE representations outperform additional CCA baselines**: Tuples (t,e) in the legend indicate the number principle components for T and E data used as input for CCA alignment. Clusters of coupled AE representations ($\lambda \in \{1, 10\}$) agree with transcriptomic class labels. (A,B) and are consistent across modalities (C)

[Meta-Review · NeurIPS 2019]

The authors present novel work regarding the ability of coupled autoencoders to analyse multi-modal profiles of neurons. The reviewers are unanimous to say that the paper is very interesting and have underlined its significance and potential impact in neuroscience. Several issues were raised by the reviewers about lack of details in the experimental part were addressed in the rebuttal. The reviewers appreciated that and agreed on acceptance. In the camera-ready version, some efforts will be needed to improve the clarity of the presentation as suggested by the reviewers.